# Prospect of Mesenchymal Stem-Cell-Conditioned Medium in the Treatment of Acute Pancreatitis: A Systematic Review

**DOI:** 10.3390/biomedicines11092343

**Published:** 2023-08-23

**Authors:** Ke Pang, Fanyi Kong, Dong Wu

**Affiliations:** 1Department of Gastroenterology, Peking Union Medical College Hospital, Chinese Academy of Medical Sciences and Peking Union Medical College, Beijing 100730, China; pangk19@student.pumc.edu.cn (K.P.); kongfy18@student.pumc.edu.cn (F.K.); 2Clinical Epidemiology Unit, Peking Union Medical College Hospital, Chinese Academy of Medical Sciences and Peking Union Medical College, Beijing 100730, China

**Keywords:** conditioned medium, acute pancreatitis, mesenchymal stem cell

## Abstract

Mesenchymal stem cells (MSCs) have demonstrated potential in both clinical and pre-clinical research for mitigating tissue damage and inflammation associated with acute pancreatitis (AP) via paracrine mechanisms. Hence, there has been a recent surge of interest among researchers in utilizing MSC cultured medium (CM) and its components for the treatment of AP, which is recognized as the primary cause of hospitalization for gastrointestinal disorders globally. A systematic review was conducted by searching the MEDLINE, EMBASE, and Web of Science databases. Studies that involve the administration of MSC-CM, extracellular vesicles/microvesicles (EVs/MVs), or exosomes to AP animal models are included. A total of six research studies, including eight experiments, were identified as relevant. The findings of this study provide evidence in favor of a beneficial impact of MSC-CM on both clinical and immunological outcomes. Nevertheless, prior to clinical trials, large animal models should be used and prolonged observation periods conducted in pre-clinical research. Challenges arise due to the lack of standardization and consensus on isolation processes, quantifications, and purity testing, making it difficult to compare reports and conduct meta-analyses in MSC-CM-based therapies.

## 1. Introduction

Acute pancreatitis (AP) is a gastrointestinal inflammatory disease characterized by injury to acinar cells and inflammation in the pancreatic tissue. It is the most common cause of hospital stays among gastrointestinal diseases [1,2]. Clinical manifestations of AP range from mild forms (80–85% of cases) to severe forms (15–20% of cases) [1,2,3].

Early phases of pancreatitis are marked by acinar cell death, edema, and histopathological changes induced by the activation and release of pancreatic enzymes. Activation of nuclear factor kappa B (NF-κB) in acinar cells also occurs, along with the production of inflammatory mediators by immune cells and injured acinar cells, exacerbating the inflammatory cascade in AP patients [4,5,6]. While treatment options such as endoscopic retrograde cholangiopancreatography (ERCP), non-steroidal anti-inflammatory drugs (NSAIDs), prophylactic antibiotics, and enteral nutrition are available, the mortality rate remains high, particularly in severe cases [1,6].

Given that current strategies primarily target symptoms rather than the root cause of the disease, there is a need for new treatment approaches for more effective management of this complex gastrointestinal disorder [4,7]. Mesenchymal stem cells (MSCs) and their secreted molecules have emerged as a potential therapeutic approach for various inflammatory and immune-mediated disorders, with researchers investigating their protective effects in treating AP since 2011 [8,9,10,11,12].

Despite the promise of MSC-based therapy, the survival ability of MSCs has been proven in many in vivo studies to be limited, making the paracrine mechanism of their cultured medium (CM) or secretome a more significant factor in mediating therapeutic effects [13,14]. CM is composed of soluble proteins, lipids, nucleic acids, and extracellular vehicles (EVs) or micro-vesicles (MVs), including exosomes and shedding vesicles [15,16]. Recent studies have reported the protective effects of CM in clinical trials and animal models, showcasing their potential in tissue repair and modulation of the microenvironment [17,18,19,20,21]. Studies have highlighted the regenerative properties of these vesicles, including their anti-apoptotic, anti-inflammatory, and angiogenic effects, as well as their ability to recruit neighboring healthy cells and stabilize the microenvironment of necrotic tissue [14].

One advantage of utilizing CM is its potential to resolve safety concerns associated with direct stem cell transplantation, such as tumorigenicity, immune incompatibility, and transmission of infections [10]. Moreover, CM can be produced in large quantities, stored for extended periods without losing potency, and modified for specific therapeutic effects, all of which can reduce the cost and time involved in cell-based therapy.

Previous in vitro studies have shown promising outcomes regarding the use of CM for treating AP. Exosomes derived from bone marrow MSCs overexpressing klotho reversed apoptosis and NF-κB activation in AP-stimulated pancreatic cancer cells [22]. Additionally, MSC-CM contained immune-related molecules, such as lysosome-associated membrane proteins, major histocompatibility complex class I and II molecules, and a large number of mRNAs, microRNAs (miRNAs) of immunoregulatory rules [16]. These molecules stimulated anti-inflammatory regulatory T cells (Treg) while inhibiting the proliferation of inflammatory CD4+ T cells, CD8+ T cells, and natural killer cells, especially in AP-induced inflammation. Furthermore, MSC secretomes demonstrated the ability to transmit biological information to target cells, modulating their activities through signaling pathways like vascular endothelial growth factor (VEGF) and Shh/Gli activation, leading to vascular protection [23,24,25]. Moreover, during the occurrence of pancreatic inflammatory diseases, segmental or diffuse inflammation of the pancreatic parenchyma can lead to pancreatic necrosis, fibrosis, atrophy, and the loss of acinar and islet cells. These structural changes result in pancreatic endocrine and exocrine dysfunction. However, exosomes have shown the capacity to stimulate cell proliferation and induce cell activation, raising speculation that they may serve as a theoretical basis for the treatment of pancreatic diseases characterized by inflammation and structural damage [26,27].

Thus, the main purpose of this review is to provide a comprehensive summary of the best available evidence from in vivo studies on the use of stem cell CM for the treatment of acute pancreatitis. By analyzing and consolidating the data, this review aims to guide future research in the field and shed light on the prospects of utilizing MSC-derived CM as a potential therapeutic approach for AP.

## 2. Materials and Methods

### 2.1. Eligibility Criteria

The inclusion criteria for this review consisted of the following:Study subjects: in vivo studies using animal models with AP;Interventions: any application of MSC-CM, EVs/MVs, or exosomes to the study groups;Outcomes: any functional, histological, physiological, and biomechanical outcomes;Study design: comparative studies.

The exclusion criteria for the preclinical studies were the absence of histological scores or data on physiologic parameters. Studies involving experiments conducted in animal models with chronic pancreatitis models were excluded.

### 2.2. Literature Search and Study Selection

The Preferred Reporting Items for Systematic Reviews and Meta-Analyses (PRISMA) were used to guide the systematic search [28], and this systematic review has been registered on the international prospective register of systemic reviews (PROSPERO, id: CRD42023450919). Articles published up until 18 July were searched using Embase, Medline, and the Web of Science. In addition, the references from pertinent review articles for any studies that the database search might have missed were manually checked. Acute pancreatitis, mesenchymal stem cells, mesenchymal stromal cells, bone marrow mesenchymal stromal cells, umbilical cord mesenchymal cells, and stem cells were the keywords used. One researcher independently evaluated the relevance of the studies found by the systematic search by reading their titles and abstracts. 

After removing duplicates and review articles, the titles and abstracts were scanned for eligibility by two authors (K.P. and F.K.) independently. Additional searches were performed using the reference lists of the previously included studies. The full text of all selected studies is read by the same authors to apply inclusion and exclusion criteria. Any disagreement between the two authors was resolved by discussion.

### 2.3. Methodological Quality Assessment and Risk of Bias

The methodological quality of the included studies was assessed using Animal Research: Reporting of In Vivo Experiments (ARRIVE) guidelines [29]. Hence, we used modified ARRIVE combined with Consolidating Reporting of Trials [30]. Internal validity was assessed using the Systematic Review Centre for Laboratory Animal Experimentation’s risk of bias tool [31]. Two authors (K.P. and F.K.) performed all the assessments independently. Any discrepancy was resolved through discussion with other authors.

### 2.4. Data Extraction

Data from the included studies were extracted by two independent authors (K.P. and F.K.) and discrepancies were resolved through discussion until consensus was reached.

The following information was then extracted: study design, type of animal used for in vivo studies, establishment of animals or cells in included studies, type and specific donor of MSCs, isolation of CM, EVs, or exosomes, preconditioning of MSCs or CM, interventions, comparison, timing of therapy start, length of follow-up, main outcome for in vivo studies, any significant deviations from control or baseline, and other outcomes are all taken into account. The establishment of the animal interventions and any follow-up would be recorded, along with any attempts at blinding, if any were made.

We examined any quantitative outcome measures comparable to clinical outcome measurements (i.e., histological ratings indicating the extent of tissue damage and serum amylase and lipase levels), and biomechanical testing as the primary endpoints for in vivo investigations. Table 1 displays the data collected for in vivo study results. Due to the significant degree of data heterogeneity (i.e., MSC source, subject animals, outcome measures, and follow-up time), no meta-analysis could be produced.

## 3. Results

### 3.1. Study Selection

Figure 1 depicts a PRISMA flow diagram that encapsulates the research selection procedure. From the literature, 2594 studies in total were found. Ninety papers were suitable for additional review after the titles and abstracts were scrutinized. Six articles (containing 8 separate comparison experiments) were included in this systematic review after full-text evaluation.

### 3.2. Study Characteristics

An overview of the study is presented in Table 1, and the specific explanation about the study design is shown in Table A1, including the methods used for MSC identification and isolation of CM or exosomes, the grouping of experiments, the timing of therapy and the length of observation.

#### 3.2.1. Donor Pretreatment and Identification of MSCs

Three studies utilized MSCs from allogeneic mice/rats [33,36,37], while others introduced human MSCs [32,34,35]. The types of MSC used were dispersed from umbilical cord- derived mesenchymal stem cells (UC-MSCs) [32], hair follicle-derived mesenchymal stem cells (HF-MSCs) [33], adipose-derived mesenchymal stem cells (AD-MSCs) [35,36], and bone marrow mesenchymal stem cells (BM-MSCs) [37], to induced stem cells (iMSCs) [34].

Abdolmohammadi et al. subjected AD-MSCs to hypoxic conditions prior to isolating the cultured medium. However, their findings revealed no statistically significant differences in either immunological or clinical results between the group that had hypoxia pretreatment and the group that was conditioned under normal conditions [36].

Flow cytometry was the predominant method used for the detection of MSCs in all six investigations. Although the specific cellular markers utilized in different research studies may vary owing to unique characteristics of different tissue origin, it is commonly observed that MSCs typically demonstrate positive expression of stem cell markers such as CD29, CD44, and CD90, while exhibiting negative expression of non-stem cell markers such as CD34 and CD45, as has been demonstrated in previous investigations [38,39]. Li et al. employed immunofluorescence labeling and Western blot techniques to better characterize HF-MSCs in their study [33].

#### 3.2.2. Isolation and Identification of MSC-CM/Exosome

Roch et al. used a 3 kDa filter to infiltrate MSC-CM before administration [35] while Abdolmohammadi et al. used a 0.22 µm filter [36]. In most investigations with MSC exosomes, the primary and exclusive method employed for separation and identification was ultracentrifugation, followed by resuspension and subsequent examination of morphology using transmission electron microscopy (TEM). Li et al. employed Western blotting to label small extracellular vesicles (sEVs) using EV markers [33]. Additionally, they quantified the protein content of the isolated EVs.

#### 3.2.3. Model Establishment and Grouping

All of the studies utilized murine models (three in rats, five in mice). No large animals were used in any of the present studies. AP was induced in animals by cerulein injection [33,35,36] or biliary retrograde injection of sodium taurocholate (NaT) [34,37]. The former usually results in pancreatic edema, while the latter causes severe or necrotizing pancreatitis, and is closer to the etiology in humans [35]. Zhao et al. established traumatic AP models by adding different grades of extrusion stress to the rats’ exposed pancreatic tissue via surgery [32]. The number of animals employed within each experimental cohort exhibits a range spanning from six to eighteen subjects. It is noteworthy to mention that none of the six studies under scrutiny have disclosed a methodology pertaining to the calculation of sample sizes. Among the eight investigations scrutinized, a majority of six exclusively incorporated ordinarily administered subjects, together with those afflicted by acute pancreatitis and treated solely with the vehicle, as their controls. Conversely, the remaining two studies introduced normally administered animals treated with MSC-CM as a supplementary control, ostensibly to facilitate the detection of any unanticipated adverse reactions elicited. Nonetheless, the empirical findings of these investigations do not report the manifestation of any such adverse effects.

#### 3.2.4. Timing of Therapy and Observation

The administration of MSC-CM or exosomes is commonly initiated promptly subsequent to the induction of AP, with the interval spanning from immediate application to a maximum delay of 2 h. Euthanasia of the subjects often takes place within a timeframe of 24 to 48 h, eschewing intermediate observations, in order to facilitate the retrieval of pancreatic tissue and hemodynamic samples for the comprehensive assessment of experimental outcomes. 

### 3.3. Methodological Quality

#### Assessment of Risk of Bias

SYRCLE’s risk of bias tool was employed to evaluate the methodological robustness of the encompassed studies. Illustrated in Figure 2 is the delineation of the risk of bias within individual studies, revealing that all six studies (comprising eight distinct experiments) merely indicated randomization without expounding on particulars. Consequently, these studies were designated as having an “unclear” risk of bias regarding sequence generation. Notably, a noteworthy concern arose from the baseline characteristics in the majority of the included studies. In these instances, animals were grouped prior to the induction of AP. However, considering the interventional nature of the studies, the induction of the disease ought to precede random allocation or post-grouping. Under these circumstances, the temporal randomness of disease induction assumes importance—an aspect unaddressed in the cited articles.

The assessment of allocation concealment resulted in an “unclear” risk of bias for studies that assigned animals to treatment and control groups without furnishing explicit details. Conversely, all studies provided explicit documentation of random housing, thus precluding a “high” risk of bias classification within the random housing domain. With unanimity, all studies underscored the blinding of participants and personnel, as well as the blinding of outcome assessors, warranting a “low” risk of bias designation.

The randomness of outcome assessment, a pivotal facet, incurred an “uncertain” risk of bias in three of the incorporated experiments due to the dearth of reported specifics. This pertained particularly to instances where histological scores were manually evaluated without elucidating participant blinding.

The scrutiny of attrition and reporting bias yielded a “low” risk of bias for the majority of the studies, thanks to comprehensive enumeration of animal numbers across both the study design and results sections. Furthermore, other dimensions of risk of bias exhibited a consistently “low” risk across all the included studies. Evaluation of this category encompassed considerations such as the absence of crossover design, contamination, and conflicts of interest, as well as the disclosure of ethical approvals.

Regrettably, none of the studies documented a priori sample size/power calculations. Summarily, the cumulative assessment of risk of bias indicates that the included studies are of moderate quality concerning their design. A comprehensive overview of the risk of bias across the ten domains of the included studies is provided in Figure A1.

### 3.4. Clinical Related Outcomes

#### 3.4.1. MSC Cultured Medium Alleviates Pancreatic Injury and Reduces Serum Pancreatic Enzyme

Seven out of eight experiments showed that histopathological scores of animal pancreas were significantly lowered after MSC cultured medium therapy. The histopathological scores usually included measurements of overall tissue necrosis, edema, hemorrhage, and inflammatory infiltration on a numeric scale, or in one study, included the percentage of acinar cell vacuolization and leukocyte infiltration [36]. The scores were evaluated by pathologists according to H&E-stained sections under a microscope following published criteria [40,41] or criteria set by the researchers. Differences between these evaluation criteria pose the greatest obstacle to conducting a meta-analysis.

#### 3.4.2. MSC Cultured Medium Lessens Myocardial Injury and Restores Cardiac Function

Chen et al. aimed to investigate the impact of MSC exosomes on cardiac rupture, a serious systemic manifestation resulting from AP. The administration of exosome therapy in AP rats resulted in notable enhancements in heart function, reductions in infarction ratio, and suppression of oxidative stress levels [34].

### 3.5. Immunological Outcomes

#### MSC Culture Medium Reduces Inflammatory Responses and Apoptotic Processes

MSC-CM elicits a reduction in inflammatory mediators, specifically interleukin-6 (IL-6), tumor necrosis factor-alpha (TNF-α), and myeloperoxidase (MPO), while concurrently augmenting the expression of anti-inflammatory indicators such as interleukin-4 (IL-4) and interleukin-10 (IL-10). Additionally, the NF-κB pathway, responsible for up-regulating the expression of genes involved in immune cell development, immune cell activation, and cytokine production, was found to be significantly suppressed by MSC MVs by Yin et al. [37]. Chen et al. showed that the administration of MSC exosomes resulted in enhanced cell viability. This impact was achieved through the activation of the Akt/nuclear factor E2-related factor 2 (Nrf2)/heme oxygenase 1 (HO-1) signaling pathway, which aligns with the observations made by Zhao et al., who also reported an apoptotic effect [32,34].

## 4. Discussion

Our primary finding was that MSC-CM reduced pancreatic injury in preclinical studies. All six studies showed improvement of clinical and immunological outcomes, but this result needs to be interpreted with caution, given the high heterogeneity of the included studies.

### 4.1. The Role of MSC Cultured Medium in Alleviating Pancreatic Injury and Suppressing Inflammation in Murine Models

The observed mitigation of pancreatic injury and reduction in serum amylase and lipase levels can be attributed to the effective suppression of both pancreatic and systemic inflammation, as supported by the immunological outcomes. Figure 3 provides a summary of the therapeutic roles of mesenchymal stem cell (MSC)-based therapy in acute inflammatory pancreatic diseases based on published studies [14,15,42,43,44]. Furthermore, Chen et al. demonstrated that MSC exosome administration enhanced cell viability through Akt/Nrf2/HO-1 signaling, corroborating Zhao et al.’s findings of an apoptotic effect. These findings collectively highlight the potential of MSC-CM in addressing systemic symptoms caused by acute pancreatitis [32,34].

None of the studies reported any side effects of MSC-CM. Additionally, only one study directly compared a CM group to an MSC whole transplant group [35]. However, this study solely assessed histopathological scores and did not find any significant difference between the two groups.

### 4.2. Challenges in MSC-CM Preparation

Various tissue sources have indeed been explored to generate populations of MSCs, as indicated by the diverse origins of MSCs in the six concluded research studies. However, isolated MSCs often exhibit heterogeneity in terms of in vitro properties (cellular markers or lineage) and functional phenotypes [10]. This heterogeneity is reflected in observable in vitro properties, such as flow cytometry and multilineage differentiation potential, as well as the in vivo functional phenotype, which has been demonstrated to vary. These variations can significantly impact the frequency of mesenchymal progenitors within the resulting populations and the therapeutic potency of their CM. Nevertheless, the studies available in Table 1 are insufficient to determine which origin of the MSC secretome holds the highest therapeutic potency. Further research is required to address this question adequately.

MSC cells usually undergo characterization by flow cytometry using established surface markers [45]. The MSCs’ capability of differentiation is also evaluated to define their stemness [13]. However, the isolation procedure for MSCs in the six studies is far from standard, and the conservation after isolation is seldom mentioned in the articles.

The method of CM storage is also under-reported, although it is one of the methodological concerns that affects both the morphology and physicochemical parameters of EVs [46,47]. Precise details for characterization of these EVs may help researchers reproduce animal-based studies [27].

### 4.3. Limitations in AP Models and Study Designs

The use of both small and large animal models has been prevalent in acute pancreatitis research; however, there is a lack of reports on the application of MSC-CM in large animals, which also applies to MSC transplantation experiments. Typically, the transition from small to large animal models is a necessary step before embarking on first-in-human clinical trials. While C5BL/6 mice have been utilized, Sprague Dawley rats have been the primary subjects in the majority of animal investigations thus far. This raises the question of how effectively a large animal model can replicate the AP induction observed in small animals.

The included studies incorporated two chemical methods for inducing AP, specifically cerulein and sodium taurocholic acid (Na-T) injection. Na-T injection is the method most frequently used to induce severe AP, also known as acute hemorrhagic pancreatitis, which closely resembles the mechanism of human AP development [37]. Conversely, cerulein is often employed when mild AP is desired [9]. Studies have concluded that Na-T-induced AP in tiny animals exhibits histological changes, pancreatic protease activation, and multi-organ dysfunction syndrome that closely resemble the human condition known as multi-organ dysfunction syndrome [48]. Among the spectrum of chemical methodologies deployed for AP induction, noteworthy alternatives encompass the Lieber-DeCarli diet, which involves the introduction of alcohol into a liquid diet, and the administration of ethanol in conjunction with fatty acids [49]. These models are designed to emulate the excesses associated with alcohol consumption in humans, rendering them clinically relevant. Notably, the ethanol plus fatty acid injection paradigm incites potential systemic toxicity, rendering it increasingly preferred, especially when considering systemic AP syndromes. Beyond the domain of chemical induction strategies, an emerging trend entails the utilization of pancreatic duct ligation models, virus-induced pancreatitis models, and models arising from endoscopic retrograde cholangiopancreatography (ERCP) induction [50]. These methodologies have garnered ascendancy in contemporary research, especially in larger animals. Given that the mere presence of necrosis within human AP does not invariably translate into adverse outcomes, a broader array of AP models necessitates consideration. This forward-looking approach underscores the importance of incorporating diverse models in future research endeavors, thereby enabling a comprehensive assessment of the efficacy of MSC-CM therapy across a spectrum of pathophysiological contexts.

Defects have also been observed in the study designs. In addition to the absence of random sequence generation, inadequate blinding of experimental participants, and an insufficient estimation of sample size, all of which are thoroughly discussed in the risk of bias assessment, the majority of studies require a prolonged observation period subsequent to the treatment of AP animals with MSC-CM. The follow-up duration in the recorded eight experiments varied between 12 and 24 h after AP induction. However, in most cases, this observation period was insufficient to fully understand the entire transport process. Additionally, the disease’s progression during the application of MSC-CM may influence its therapeutic impact. Although MSC and its paracrine factors have shown promise in treating chronic inflammatory disorders [51,52], their effectiveness in acute pancreatitis remains unexplored in the current study. Furthermore, most studies did not assess the biodistribution of administered EVs, which could provide valuable insights into EV homing in various organs.

The studies used a range of CM or EV dosages (20–1000 μg) and quantified EVs through protein content estimation assays. However, protein- or particle-based quantifications may not indicate impurities, and cell numbers may not reflect their physiological status. Isolation methods can impact EV quality and co-isolated non-EV materials, affecting the administered dose’s consistency and efficacy. Standardized approaches are essential for accurate dosing and reliable therapeutic outcomes. Therefore, the minimal information for studies of extracellular vesicles (MISEV2018) guidelines strongly suggest the use of more than one parameter [53].

### 4.4. Future Directions

Undoubtedly, given the absence of MSC utilization in comprehensive animal AP models, this avenue stands out as a crucial focus for future research endeavors. Additionally, there is a pressing need for more in-depth investigations into the immunoregulative mechanism of both MSCs and MSC-CM.

Pre-conditioning of MSC-CM and editing of EVs have shown promising results in enhancing therapeutic potential [22]. Abdolmohammadi et al. discovered that HP-MSCs exhibited significantly higher levels of total protein expression; however, no significant differences in clinically relevant outcomes were observed [36]. Nevertheless, pursuing this approach is still worthwhile, as modifications to MSCs, as demonstrated in other disease models, have proven effective [54]. These modifications include: (a) cultivating cells under inflammatory conditions to amplify the production of growth factors and anti-inflammatory molecules; (b) employing pro-inflammatory stimuli to induce greater secretion of immune-related factors; (c) nurturing tri-dimensional growth to augment the production of anti-tumoral and anti-inflammatory factors; and (d) employing microparticle engineering.

## 5. Conclusions

The review concludes that MSC-conditioned medium shows effectiveness in reducing pancreatic injury in preclinical studies. However, prior to initiating clinical trials, it is essential to employ large animal models and conduct prolonged observation periods in pre-clinical research. A lack of standardization and consensus on various aspects of isolation processes, quantifications, and purity testing poses significant challenges in MSC-CM-based therapies, making it difficult to compare reports and conduct meta-analyses. Additionally, the mechanism of action of CM and EVs, crucial for translating preclinical data to clinical applications, still needs further investigation and determination.

## Figures and Tables

**Figure 1 biomedicines-11-02343-f001:**
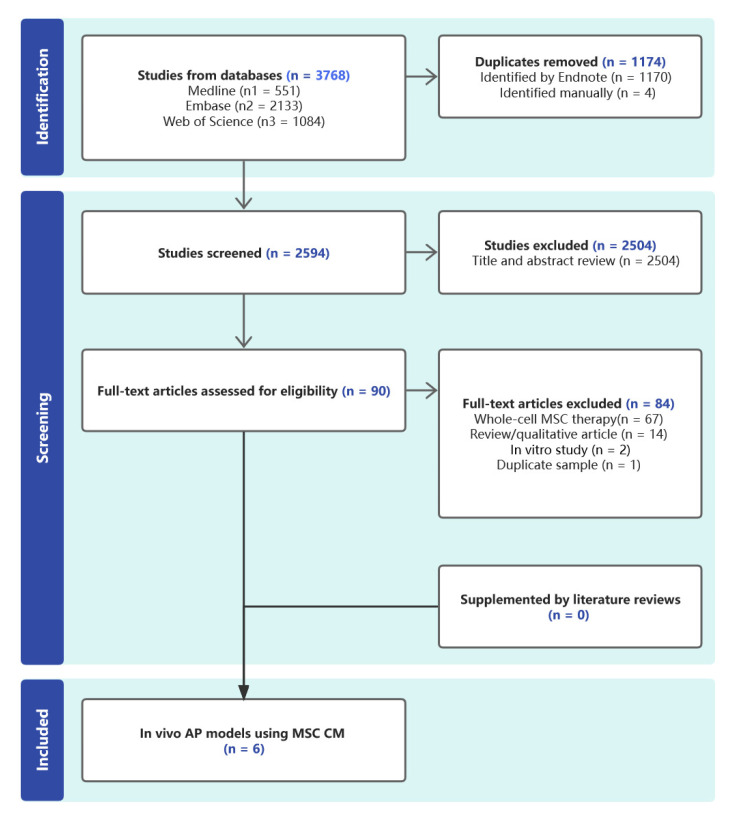
A flow chart displaying the studies that qualify for the review.

**Figure 2 biomedicines-11-02343-f002:**
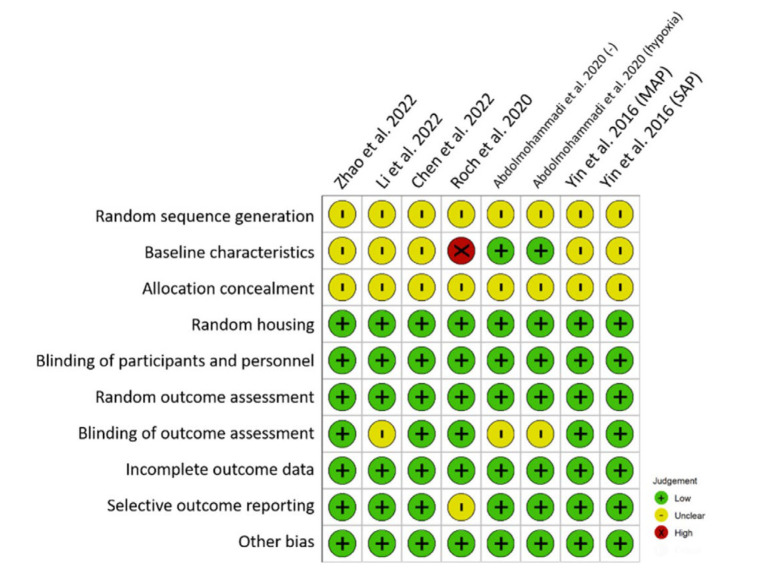
Risk of bias assessment in individual studies [32,33,34,35,36,37].

**Figure 3 biomedicines-11-02343-f003:**
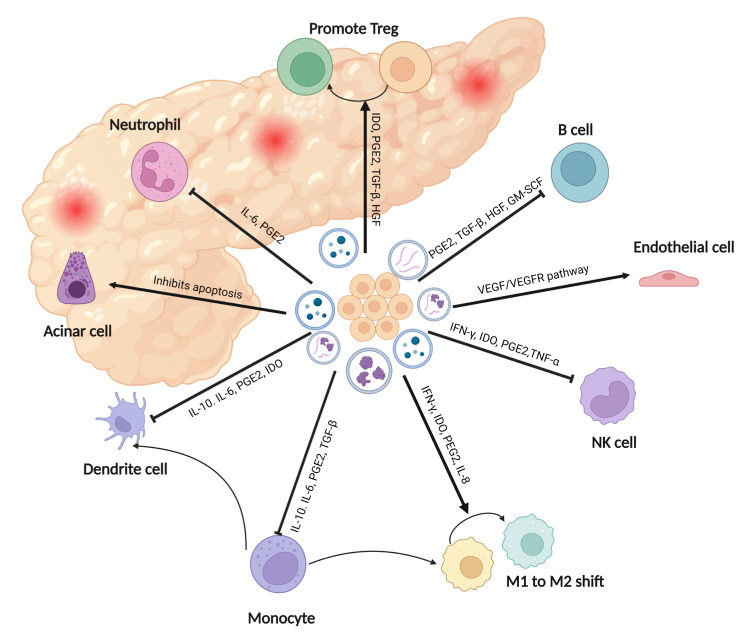
The therapeutic roles of MSC-based therapy in inflammatory pancreatic diseases. Created with BioRender.com.

**Table 1 biomedicines-11-02343-t001:** Overview of the studies.

Authors/Year/Ref	Donor/Type/Pre-Treatment	Level of CM	Model/Induction Method/Severity of AP	Routs/Dose/Number of Doses Mouse: 20 g Rat: 250 g	Immunological Outcome	Clinical Outcome	Level of Evidence
Zhao et al. (2022) [32]	Human/UC-MSC/-	Exo	Rat/extrusion stress/traumatic AP	Tail vein/20 μg/single	↓ Apoptosis	↓ Histopathological scores ↓ Serum amylase & lipase	Moderate
Li et al. (2022) [33]	Mice/HF-MSC/-	sEV	Mice/cerulein/AP	Intraperitoneal and tail vein/100 μg/double	↓ TNF-α ↓ IL-6 ↑ IL-4 ↑ IL-10 ↓ MPO	↓ Histopathological scores ↓ Serum amylase & lipase	Moderate
Chen et al. (2022) [34]	Human/iMSCs/-	Exo	Rat/NaT/AP	Tail vein/100 μg/single	↑ Akt/Nrf2/HO-1 ↑ vWF and VEGF	↓ Myocardial injury ↓ Oxidative stress ↑ Cardiac function	Moderate
Roch et al. (2020) [35]	Human/AD-MSC	CM	Mice/cerulein/AP	Tail vein/100 μL/single	-	↓ Histopathological scores	Low
Abdolmohammadi et al. (2020) [36]	Mice/AD-MSC/-	CM	Mice/cerulein/AP	Tail vein/500 μL/triple	↓ IL-6 ↓ MPO	↓ Histopathological scores ↓ Serum amylase & lipase	Moderate
Abdolmohammadi et al. (2020) [36]	Mice/AD-MSC/hypoxia-preconditioned	CM	Mice/cerulein/AP	Intraperitoneal/500 μL/triple	↓ IL-6 ↓ MPO No significant difference with CM	↓ Histopathological scores ↓ Serum amylase & lipase No significant difference with CM	Moderate
Yin et al. (2016) [37]	Rat/BM-MSC/-	MV	Rat/NaT/SAP	Tail vein/1000 μg/single	↓ NF-κB, p65 expression ↓ MPO ↑ Acinar cells survival	↓ Histopathological scores ↓ Serum amylase & lipase	Moderate
Yin et al. (2016) [37]	Rat/BM-MSC/-	MV	Mice/Cerulein/MAP	Tail vein/100 μg/single	↓ NF-κB, p65 expression ↓ MPO ↑ Acinar cells survival	↓ Histopathological scores ↓ Serum amylase & lipase	Moderate

↓/↑ Indicates a significant decrease/increase as compared with vehicle-treated control. Abbreviations: Regarding MSCs: UC-MSC, umbilical cord-derived stem cells; HF-MSC, hair follicle-derived stem cells; iMSC, induced stem cells; AD-MSC, adipose-derived mesenchymal stem cells; BM-MSC, bone marrow mesenchymal stem cells. Related to preparation procedures: CM, conditioned medium; Exo, exosome; sEV, small extracellular vesicles; MV, micro-vesicles. Any study that exhibits a high-risk domain is categorized as having a low level of evidence, while studies with low or unclear risk domains are designated as having a moderate level of evidence.

## Data Availability

The data underlying this article are available in the article.

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
