# Peer review of "Prospect of Mesenchymal Stem-Cell-Conditioned Medium in the Treatment of Acute Pancreatitis: A Systematic Review"

_biomedicines, 2023, doi:10.3390/biomedicines11092343_

Round 1

Reviewer 1 Report

This is a very well-written and comprehensive review. Therefore, there are only some minor issues to be corrected in the manuscript:

1. In the title of the manuscript (line 3) the word "systemic" should be replaced by the word "systematic"

2. The same should be done in the line 17

3. Line 241: Is IL-10 an inflammatory mediator? Please, explain!

4. In Figure 3. the shortages PEG2 (3x) should be corrected into PGE2.

Minor editing of English language required.

Author Response

Dear reviewer,

Thank you for your thoughtful review of our manuscript. This article represents the first author's first experience in submitting a manuscript, so we genuinely appreciate your positive feedback and the chance to address the concerns you've raised:

1. In the title of the manuscript (line 3) the word "systemic" should be replaced by the word "systematic".

Author response: We apologize for the oversight and thank you for bringing this to our attention. The word "systemic" has been replaced with "systematic" in the title of the manuscript, as well as in line 17.

2. The same should be done in the line 17

Author response: Once again, thank you.

3. Line 241: Is IL-10 an inflammatory mediator? Please, explain!

Author response: We apologize for any confusion caused. In fact, IL-10 is generally considered an anti-inflammatory cytokine, playing a crucial role in suppressing inflammatory responses. We have revised the text in line 263 to clarify this point: "MSC CM elicits a reduction in inflammatory mediators, specifically interleukin-6 (IL-6), tumor necrosis factor-alpha (TNF-α), and myeloperoxidase (MPO), while concurrently augmenting the expression of anti-inflammatory indicators such as interleukin-4 (IL-4) and interleukin-10 (IL-10)."

4. In Figure 3. the shortages PEG2 (3x) should be corrected into PGE2.

Author response:  The typographical error in Figure 3 has been corrected.  Many thanks.

We trust these revisions address your concerns and enhance the manuscript's quality. Your feedback is vital in improving accuracy and clarity; thank you for your contribution.

Reviewer 2 Report

This is a systemic review conducted by searching the MEDLINE, EMBASE, and Web of Science. Any study that involves administration of MSC CM, extracellular vesicles/microvesicles (EVs/MVs), or exosomes to acute pancreatitis animal models are included. A total of six research studies, including eight experiments, were identified as relevant. The findings of this study provide evidence in favour of a beneficial impact of MSC CM on both clinical and immunological outcomes. The authors concluded that prior to clinical trials, large animal models should be used, and prolonged observation periods conducted in pre-clinical research. Challenges arise due to the lack of standardization and consensus on isolation processes, quantifications, and purity testing, making it difficult to compare reports and conduct meta-analyses in MSC CM-based therapies.

This is a well-structured review demonstrating some important conclusions. Limitations of the administration of MSC CM in acute pancreatitis are correctly discussed.

Comments:

The main limitation of the review is the low number of research studies involved. Is it enough to get the sufficient information about the topic?

AP was induced in animals by cerulein injection or biliary retrograde injection of sodium taurocholate (NaT). The former usually results in pancreatic edema, while the latter causes severe or necrotizing pancreatitis, and is closer to the etiology of human. Zhao et al. established traumatic AP models by adding different grade of extrusion stress to the Rats’ exposed pancreatic tissue by surgery. Although the biliary retrograde injection of NaT induced form can mimic the biliary pancreatitis, the pathomechanism of several other forms are completely different, i.e. hypertriglyceridemia-induced necrotizing pancreatitis viral infection provoked less severe forms. Since there are not animal models for these forms, the use of MSC CM in these cases is questionable. It should be discussed.

Line 187. A sentence seems to be unfinished.

Author Response

Dear reviewer,

Thank you for your thoughtful review of our manuscript. We genuinely appreciate your positive feedback and the chance to address the concerns you've raised:

1. The main limitation of the review is the low number of research studies involved. Is it enough to get the sufficient information about the topic?

Author response: I completely share your view. While it is true that the number of included research studies in our review is limited, we wish to emphasize that we have diligently tracked this topic since February 2022. As of now, we believe that the studies we have included encompass nearly all, if not all, of the pertinent published research concerning the application of MSC secretome in acute pancreatitis. We chose to publish the literature research findings because the first clinical trial of MSC whole cell transplantation in acute pancreatitis, demonstrating effective immunoregulation, was published in June of this year, and we'd like to at least offer valuable insights to fellow researchers who share our objective of putting MSC conditioned media into application.

2. AP was induced in animals by cerulein injection or biliary retrograde injection of sodium taurocholate (NaT). The former usually results in pancreatic edema, while the latter causes severe or necrotizing pancreatitis, and is closer to the etiology of human. Zhao et al. established traumatic AP models by adding different grade of extrusion stress to the Rats’ exposed pancreatic tissue by surgery. Although the biliary retrograde injection of NaT induced form can mimic the biliary pancreatitis, the pathomechanism of several other forms are completely different, i.e. hypertriglyceridemia-induced necrotizing pancreatitis viral infection provoked less severe forms. Since there are not animal models for these forms, the use of MSC CM in these cases is questionable. It should be discussed.

Author response:  We think this is an excellent suggestion. And we have now further discussed the AP models in Discussion, and the details are as follows (Discussion 4.3 line 323):

'The studies included incorporated two chemical methods for inducing AP, specifically cerulein and sodium taurocholic acid (Na-T) injection. Na-T injection is the method most frequently used to induce severe AP, also known as acute hemorrhagic pancreatitis, which closely resembles the mechanism of human AP development[37]. Conversely, cerulein is often employed when mild AP is desired[9]. Studies have concluded that Na-T-induced AP in tiny animals exhibits histological changes, pancreatic protease activation, and multi-organ dysfunction syndrome that closely resemble the human condition known as multi-organ dysfunction syndrome[45]. Among the spectrum of chemical methodologies deployed for AP induction, noteworthy alternatives encompass the Lieber-DeCarli diet involving the introduction of alcohol into a liquid diet, and the administration of ethanol in conjunction with fatty acids[46]. These models are designed to emulate the excesses associated with alcohol consumption in humans, rendering them clinically relevant. Notably, the ethanol plus fatty acid injection paradigm incites potential systemic toxicity, rendering it increasingly preferred, especially when considering systemic AP syndromes. Beyond the domain of chemical induction strategies, an emerging trend entails the utilization of pancreatic duct ligation models, virus-induced pancreatitis models, and models arising from endoscopic retrograde cholangiopancreatography (ERCP) induction[47]. These methodologies have garnered ascendancy in contemporary research especially in larger animals. Given that the mere presence of necrosis within human AP does not invariably translate into adverse outcomes, a broader array of AP models necessitates consideration. This forward-looking approach underscores the importance of incorporating diverse models in future research endeavors, thereby enabling a comprehensive assessment of the efficacy of MSC CM therapy across a spectrum of pathophysiological contexts.'

The authors possesses a modest understanding of constructing AP models and are appreciative of receiving additional insights and guidance.

3. Line 187. A sentence seems to be unfinished.

Author response:  Thank you for pointing this out. The previously incomplete paragraph has now been duly finalized (Line 191).

We believe that these revisions effectively tackle your concerns and elevate the overall quality of the manuscript. This article actually marks the initial submission experience for the first author. Thus, we extend our sincere gratitude for your valuable contributions.

Reviewer 3 Report

This is a very interesting systematic review that provide evidence in favour of a beneficial impact of MSC CM on both clinical and immunological outcomes

The paper is novel and it is the first ones in literature. 

I suggest to improved the summary table including the following elements

1) level of evidence

During the results description, please create sub chapters based on the level of evidence in vitro, in vivo and in humans. 

Provide future prosepective based on the results in hymans

low level

Author Response

Dear reviewer,

Thank you for your thoughtful review of our manuscript. We genuinely appreciate your positive feedback and the chance to address the concerns you've raised:

1. I suggest to improved the summary table including the following elements 1) level of evidence

Author response: We think this is an excellent suggestion. As the GRADE system is often used to evaluate the level of evidence in RCTs, we're doing this systemic reviews on purely animal models, which is applicable to SYstematic Review Centre for Laboratory animal Experimentation (SYRCLE) risk of bias tool. The comprehensive assessment of risk of bias in individual studies is presented in Figure 2. Additionally, we have introduced a new column in Table 2 to document the overall evidence level of each study.

2. During the results description, please create sub chapters based on the level of evidence in vitro, in vivo and in humans. Provide future prosepective based on the results in humans

Author response:  Thank you for this suggestion. It would indeed be valuable if we could locate any human research related to MSC CM therapies in the context of acute pancreatitis.

While it is true that the number of included research studies in our review is limited and no clinical trial was included, we wish to emphasize that we have diligently tracked this topic since February 2022. As of now, we believe that the studies we have included encompass nearly all, if not all, of the pertinent published research concerning the application of MSC secretome in acute pancreatitis. We chose to publish the literature research findings because the first clinical trial of MSC whole cell transplantation in acute pancreatitis, demonstrating effective immunoregulation, was published in June of this year, and we'd like to at least offer valuable insights to fellow researchers who share our objective of putting MSC conditioned medium into application.

It appears that our Discussion section may have posed challenges in terms of comprehensibility, and for this, we extend our apologies. We have introduced subheadings to better structure the content, aiming to create a more organized and accessible flow of information. However, it's clear that there is room for further improvement.

Besides, we understand the importance of improving its English quality. After the review, professionals will be engaged to refine the language. It's actually the lead author's first attempt in submitting a manuscript, therefore, your insights will undoubtedly contribute significantly to enhancing the quality of our work. Thank you very much for your contribution

Ke Pang